# Gut microbiota signatures of the three Mexican primate species, including hybrid populations

**Diego Zubillaga-Martín**[1,2]*, **Brenda Solórzano-García**[3], **Alfredo Yanez-Montalvo**[4¤a], **Arit de León-Lorenzana**[4¤b], **Luisa I. Falcón**[4], **Ella Vázquez-Domínguez**[1]*

**1** Laboratorio de Genética y Ecología, Departamento de Ecología de la Biodiversidad, Instituto de Ecología, Universidad Nacional Autónoma de México, Ciudad de México, México **2** Posgrado en Ciencias Biológicas, Universidad Nacional Autónoma de México, Ciudad de México, México **3** Laboratorio de Parasitología y Medicina de la Conservación, ENES-Mérida U.N.A.M., Ucú, Yucatán, México **4** Laboratorio de Ecología Bacteriana, Instituto de Ecología, Unidad Mérida, Universidad Nacional Autónoma de México, Ucú, Yucatán, México

¤ a Current address: Facultad de Ciencias, Universidad Autónoma de Baja California, Carretera Transpeninsular 3917, Fraccionamiento Playitas, Ensenada! Baja California, 22860, México
¤ b Current address: Universidad Intercultural Maya de Quintana Roo-CONAHCYT, Carretera Muna-Felipe Carrillo Puerto km 137, La Presumida, José María Morelos, 77870, Quintana Roo, México
* evazquez@ecologia.unam.mx (EVD), dzubillaga@iecologia.unam.mx (DZM)

## Abstract

Diversity of the gut microbiota has proven to be related with host physiology, health and behavior, influencing host ecology and evolution. Gut microbial community relationships often recapitulate primate phylogeny, suggesting phylosymbiotic associations. Howler monkeys (*Alouatta*) have been a model for the study of host-gut microbiota relationships, showing the influence of different host related and environmental factors. Differences in life-history traits and feeding behavior with other atelids, like spider monkeys, may reveal distinct patterns of bacterial gut communities, yet few wild populations have been studied; likewise, gut microbiota studies of hybrid populations are mostly lacking. We analyzed diversity and abundance patterns of the gut microbiota of wild populations of the three Mexican primates *Ateles geoffroyi*, *Alouatta palliata* and *A. pigra* from different regions across its distribution in the country, including sympatric localities and the *Alouatta* hybrid zone. Interspecific differences in gut microbial diversity were higher than intraspecific differences, concordant with phylosymbiosis. *Ateles* harbored the more differentiated diversity with a major presence of rare taxa, while differences were less strong between *Alouatta* species. Hybrids had a microbial diversity in-between their parental species, yet also showing unique microbe taxa. Genetic distances between *Alouatta* individuals correlated positively with their gut microbial dissimilarities. Results show that interspecific and intraspecific overall diversity, abundance and composition patterns are affected by environment, geographic distribution and host genetics. Our study provides the first comprehensive study of gut microbiota of the three Mexican primates and hybrid populations.

**Data availability statement:** The 16S raw sequence data are publicly available in NCBI under the Bioproject ID PRJNA1189536. Metadata and methods for analyses are available in the main text and in the Supporting Information. Additionally, we do not provide sampling locality coordinates because the three primate species are critically endangered, and are illegally trafficked.

**Funding:** This project was funded by Programa de Apoyo a Proyectos de Investigación e Innovación Tecnológica-Dirección General de Asuntos del Personal Académico (PAPIIT-DGAPA, UNAM; Projects IN202122 and IN202819) awarded to E.V.D. The funders had no role in study design, data collection and analysis, decision to publish, or preparation of the manuscript.

**Competing interests:** The authors have declared that no competing interests exist.

## Introduction

The gut microbiota is a community of microorganisms composed of virus, bacteria, archaea, fungi and protozoa. In mammals, it plays a crucial role in host nutrition, physiology and overall health, strongly influencing host behavior, ecology and evolution [1,2]. Likewise, host's inherent factors influence the host-microbiota dynamics across time and space (e.g., host genetics and diet; [3,4]). Bacteria are the most studied organisms, which is not surprising considering that the collective genome of bacteria in human bodies has more genes than the human genome [5], and in the colon there are more bacteria than host tissue cells [6]. Amongst the main roles that gut microbial communities play in mammals is their effect on host feeding plasticity, enabling dietary adaptations [3,7]. Heterotrophic bacteria in the gut carry out key functions to process food, especially the fermentative breakdown of short-chain fatty acids into compounds that animals can digest [8,9].

Host phylogeny can also be correlated with the gut microbiota [10,11]. The significant microbial community relationship that recapitulates host phylogeny is defined as phylosymbiosis [12]. Such associations may arise from stochastic and/or deterministic evolutionary and ecological forces, thus vertical transmission of microbes (i.e., from parents to offspring) is not obligatory since environmental acquisition can establish the same taxonomic or functional host-microbe associations [12,13]. Although phylosymbiosis is less prevalent in taxonomically richer gut microbiotas, mammals seem to be the exception as they often exhibit these associations despite hosting rich microbial communities [14]. Apparently, the host's evolutionary history largely determines which bacteria clades are found in the mammalian gut, while the feeding behavior may regulate which clades are predominant in the community [15], with herbivores having richer communities than omnivores and carnivores [10].

Non-human primates (NHP), and particularly howler monkeys (genus *Alouatta*), have been ideal systems to study host-microbiota associations, mainly due to their wide distribution across different ecosystems, encompassing diverse habitats, diets and social systems [16–18]. When contrasting the gut microbiota between different primate species, interspecific variation is consistently higher than intraspecific variation, evidencing phylosymbiosis, as well as primate-bacteria cospeciation in hominids and cercopithecids [19–21]. Similarly, microbial community divergence mostly follows expected phylogenetic relationships between Old World monkeys, New World monkeys, apes and lemurs [19,22]. However, functional convergence has been identified among folivores [22]. Additionally, other factors like environment and geography, and species-specific processes like host physiology, social group membership, as well as captivity, can strongly shape the gut microbiota in NHP [1,18,22–26]

Likewise, multiple factors affect variation of NHP gut microbial communities at the intraspecific level. For example, geographic distance is often associated to isolated host populations exhibiting distinct gut microbiota, driven by microbial demographic stochasticity and the degree of dispersal among populations of microbes and their hosts [27]. Differences in host behavior also affect microbial composition, including social group identity, social interactions and dispersal between groups, with examples in wild baboons, colobus monkeys and sifakas [28–30]. Additionally, several examples of intraspecific variation determined by environmental variables (e.g., soil, seasonality, temperature, rainfall) have been found in wild baboons [31], macaques [32], sifaka lemurs [33] and geladas [34]. Also, diet and anthropogenic disturbances have shown to influence howler monkeys and Malagasy lemurs [35,36]. Finally, intraspecific differences in gut microbiota have been linked to host genetics as well as individual sex and age [37,38]. Sex and reproductive state are known to affect gut microbiota in capuchins monkeys [39], macaques [32] and lemurs [37]. A significant effect of host genetics was observed in the Northern muriqui (*Brachyteles hypoxanthus*), where more genetically

distinct individuals had a more differentiated gut microbiota structure and composition [40], while in other studies the effect has been small [28,38].

Three NHP species are distributed in Mexico, the black and the mantled howler monkeys, *Alouatta pigra* and *A. palliata*, respectively, and the spider monkey *Ateles geoffroyi*, representing the northernmost distribution of wild primates in America. *Alouatta* monkeys have the ability to tolerate low quality diets consisting mostly of leaves during some periods of the year [41], and to maintain populations in sites with moderate anthropogenic disturbance [42]. Howlers are estimated to gain up to 31% of their daily energy from short-chain fatty acids produced by their gut microbiota [43]. Their daily nutritional demands for growth and reproduction are expected to be higher [17,44] than, for example, *Ateles* spp which consume mainly fruits [41]. *Alouatta pigra* is the most studied howler monkey species in terms of host-gut microbiota interactions. Main findings can be summarized as differences in gut microbiota observed in juvenile and female individuals compared to adults and males, respectively, and among social groups; a decrease in diversity in degraded and captive habitats has also been documented [44–48]. Seasonal changes in diet, environmental stress and parasites influence the structure of their gut microbial communities as well [7,35,49,50]. The microbiome research in these howlers has been performed in a small part of their ample distribution.

Much less has been investigated regarding the gut microbiota in *A. palliata* and, to our knowledge, no studies have been conducted to describe gut microbial patterns in free-ranging spider monkeys. Amato et al. [51] evaluated the gut microbiota variation in the two Mexican howler species and found that host species had the strongest effect, followed by forest type (evergreen and semideciduous tropical). However, *A. pigra* exhibited less microbial diversity than *A. palliata* (the latter considering populations from Nicaragua and Costa Rica) irrespective of forest type. Authors suggest that *A. pigra*'s gut microbiota is more sensitive to environmental perturbation and the associated changes in diet compared to its sister species. Additionally, Clayton et al. [23] showed that captivity is associated with loss of native gut microbial taxa in *A. palliata*. Finally, although differences between spider and howler monkeys that are assumed to affect the microbiome have been noted, the gut microbiota studies in *Ateles* have been done only with captive populations [52,53].

Hybrid populations of *A. palliata* and *A. pigra* occur in a sympatric zone in southern México [54]. Hybridization is known to cause changes in the microbiome composition and/or function in animals [13,55], but little has been studied with NHP nor with hybrid atelids. In wild baboon populations spanning a natural hybrid zone, Grieneisen et al. [56] did not find evidence of phylosymbiosis or effect of genetic distance, whereas the environment (soil) was the best predictor of variation in their gut microbiota. Host species significantly influenced alpha gut microbial diversity but not community structure in wild and captive marmosets (*Callithrix*) across four species and their hybrids [25]. Hence findings are varied and, as Miller et al. [13] emphasized, there is yet much to investigate regarding how hybridization affects the microbiome and how do hybrid organisms differ from their parentals.

Here, we analyzed a set of 40 individuals of wild black howler (*Alouatta pigra*), mantled howler (*Alouatta palliata*) and spider (*Ateles geoffroyi*) monkeys, encompassing a wide range of sites within their distribution, as well as from the *Alouatta* sympatric hybridization zone. We aimed to (1) describe the gut microbiota abundance and diversity patterns in wild populations of the three Mexican primate species; (2) evaluate the effects of phylogeny and geographic region as potential drivers of host microbial composition; and (3) assess the overall patterns of the gut microbiota in hybrid individuals and how distinct it is from the parental species. Considering that spider and howler monkeys diverged approximately 15–16 million years ago (Mya) [57,58], and black and mantled howlers diverged much more recently (3 Mya; [57]), we hypothesize that differences in gut microbial communities at the interspecific level

will be significantly higher between *A. geoffroyi* compared with the howler species (showing phylosymbiosis), while geographic region and vegetation type will also contribute to diversity differences. We expect distinct gut microbiota community structure and composition in hybrid individuals, which will harbor shared but also exclusive microbial taxa. At the intraspecific level, we predict that host geographic isolation and host genetics will be associated with distinct gut microbiota.

## Materials and methods

### Sample collection

Non-invasive sampling was performed in wild populations of the three species of Mexican primates during the dry season between 2018 and 2020, encompassing seven regions and 17 localities across their distribution in the country, plus three locations from a sympatry zone where hybrid populations of the two *Alouatta* species are known to occur [54] (Fig 1). Fresh fecal samples were collected right after deposition using gloves and a tongue depressor to avoid contamination. Samples were preserved in parafilm-sealed plastic vials containing solid NaCl at room temperature during field work and afterwards at -20°C or at -80°C for longer storage until further processing [59]. For a detailed description of the field work and sampling protocols refer to Solórzano et al. [60]; no direct handling of any individual was performed thus no ethical approval is required and fecal sampling was performed with the collecting permit from the Secretaría del Medio Ambiente y Recursos Naturales to E.V.D. (Semarnat-FAUT 0168). We classified the vegetation type of the sampling sites as tall evergreen forest (TEF), semi-deciduous forest (SDF), medium evergreen forest (MEF), and mature secondary vegetation (MSV), based on [61] (S1 Table). We selected 40 samples for this study: 11 samples from *Alouatta palliata*, 10 from *A. pigra*, nine from *Alouatta* hybrids and 10 from *Ateles geoffroyi* (S1 Table). Hybrid individuals were confirmed by population genetics and introgression analyses based on seven microsatellite loci (Vázquez-Domínguez et al. [Unpublished]).

### DNA extraction and 16S rRNA amplification

DNA extractions of fecal samples (~200 mg) were done using the QIAamp® Fast DNA Stool Mini Kit (Qiagen) following the manufacturer's instructions. Amplification of the 16S rRNA V4 region was performed following an established protocol [62]. Each sample was amplified in three independent PCR reactions with the following conditions: 95°C for 30 s followed by 35 cycles at 95°C for 30 s, 50°C for 40 s, and 72°C for 90 s, with a final elongation step of 12 min at 72°C. PCR products were pooled and purified with Agencourt AMPure XP bead-based reagent (Beckman Coulter™). The purified amplicon library was quantified with a QUBIT fluorometer (Invitrogen). The final 20 ng/µl sample amplicon library was sequenced on an Illumina MiSeq 2x300 platform at the Yale Center for Genome Analysis, CT, USA.

### Bioinformatic analyses of the 16S rDNA V4 sequences

We demultiplexed and denoised sequences, removed chimera and singletons, assigned sequences into ASVs (Amplicon Sequence Variants) with QIIME2 v.2023.2 [63], and truncated them at position 210 with DADA2, using the plugin *qiime dada2 denoised-paired* [64]. We resolved the taxonomy using the SILVA database (release 138-99% OTUs, 515-806 region), with the *feature-classifier classify-consensus-vsearch* plugin [65]. The sequences alignment and the phylogenetic tree were done with the MAFFT method [66] and with FastTree [67], respectively.

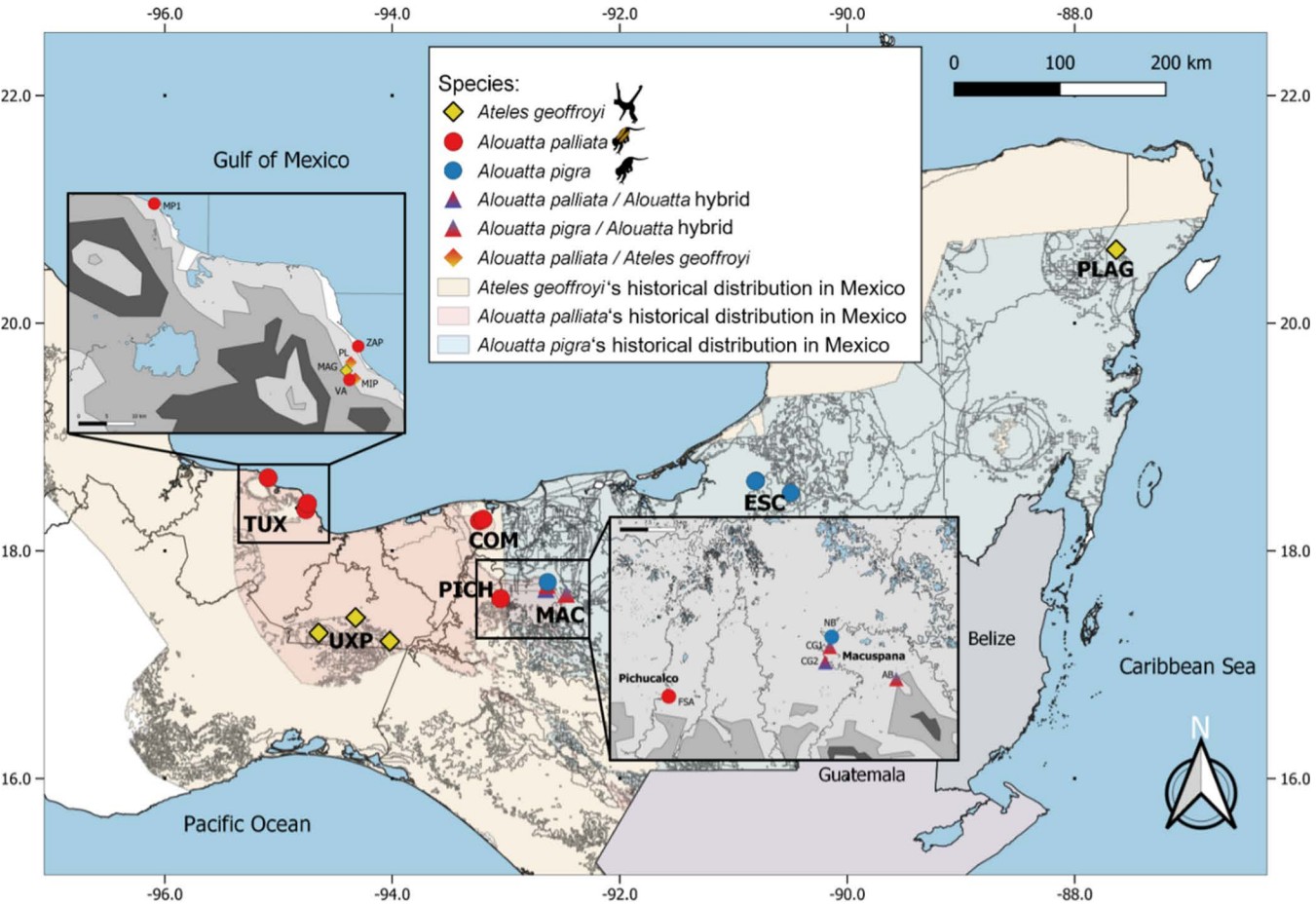

**Fig 1. Map with the localities of the samples used in this study from wild populations of the three Mexican primate species.** Spider monkeys (*Ateles geoffroyi*) localities are indicated with yellow rhomboids, mantled howler monkeys (*Alouatta palliata*) with red circles, black howler monkeys (*A. pigra*) with blue circles, and hybrids (*A. palliata* x *A. pigra*) with blue/red triangles; localities where we sampled both *A. palliata* and *A. geoffroyi* are shown with yellow/red rhomboids. The inserts respectively depict the sampling sites in Los Tuxtlas region (top left) and in the howler's hybrid zone (bottom right). Historical distributions of the three species are projected in different colors [104]. Information of geostatistical boundaries was obtained from a public domain source [http://www.conabio.gob.mx/informacion/gis/] and the map was created by us using QGIS 3.22.10 software. Names of regions are shown with black letters: TUX: Los Tuxtlas, UXP: Uxpanapa, PICH: Pichucalco, COM: Comalcalco, MAC: Macuspana, ESC: Escárcega, PLAG: Punta Laguna. Localities in Los Tuxtlas region: MP1: Montepío, ZAP: Zapoapan, PL: Playa, MAG: Magallanes, VA: Valentina, MIP: Mirador Pilapa. Localities in the hybrid region: NB: Nicolás Bravo, CG1: Carlos Green 1, CG2: Carlos Green 2, AB: Agua Blanca, FSA: Finca Santa Ana (see S1 Table). Black monkey silhouettes obtained from the open source, freely available at https://www.phylopic.org/images/aceb287d-84cf-46f1-868c-4797c4ac54a8/ateles-fusciceps; and https://www.phylopic.org/images/564c9708-bedb-4c1c-afc7-307f416901f0/alouatta-caraya.

We used the abundance tables, taxonomy classification and phylogenetic tree files created with QIIME2 for all further analyses, which were performed using different packages in the R environment v.4.2.2 [68]. The initial protocol was done with phyloseq [69] and included removal of potential synthetic samples, samples with less than 1000 reads, and Phyla with a prevalence of less than two samples. We estimated accumulation curves by host species to confirm if samples reached an asymptote; chloroplast, archaea, mitochondria and unclassified ASVs were also removed (hereafter 'filtered dataset'). We also did rarefaction without sample replacement using a rarefaction depth of 90% of the minimum sample depth (28,883.7 reads per sample) ('rarefied dataset'). Phylodiversity analyses were performed with the filtered dataset, while we used a non-filtered dataset for abundance analyses and to evaluate the complete potential diversity.

We visualized prokaryote diversity by plotting the relative abundances of the most abundant phyla (>2%) and genera (>5%) per sample, grouped by each of the host species and hybrids. We did the same for the two most abundant Bacteria phyla (Bacillota and Bacteroidota). The Bacteroidota diversity was plotted by most abundant species (>5%), while Bacillota diversity was done by most abundant families (>5%) because deeper taxonomic levels were too diverse to visualize in a graph. To visualize the number of taxa shared between primate species and hybrids we created Venn diagrams with the DrawVenn tool available online (https://bioinformatics.psb.ugent.be/webtools/Venn/) for overall diversity, for Archaea, and for the most abundant phyla of bacteria.

We did a principal coordinate analysis (PCoA) based on unweighted and weighted Unifrac distance dissimilarities, respectively, to assess prokaryote community structure among the primate species and hybrids and to visualize abundance effects. Unifrac is a powerful multivariate technique for comparing microbial communities in a phylogenetic context; unweighted takes into account only presence/absence of the taxa while weighted also considers their abundances [70]. Differences between host species were evaluated with a Permanova test [71] with the ADONIS function in vegan v.2.6.6 [72]. For comparative purposes we performed the same PCoA analysis using the rarefied dataset. We also performed an unweighted Unifrac PCoA to specifically compare individuals from the two 'parental' *Alouatta* species and the hybrids.

We quantified diversity and phylodiversity using Hill numbers [73], a method that measures diversity in units of effective ASVs (or effective lineages for phylodiversity). A key advantage of this method is that it allowed us to modulate the sensitivity towards abundant and rare ASVs (or lineages) by modifying a single parameter (q). As the value of q increases, greater weight is given to abundance, where q = 0 determines richness or number of lineages, q = 1 abundant taxa/lineages and q = 2 dominant taxa/lineages [74]. Thus, both gamma diversity and gamma phylodiversity profiles were calculated using the hilldiv package with q values from 0 to 2 [74]. Next, using hilldiv we evaluated statistical differences between host species for q = 0 and q = 1 with Kruskal-Wallis based on a Dunn test and the Benjamini-Hochberg correction, and for q = 2 with an ANOVA with Tukey post-hoc test. The phylogenetic tree used for phylodiversity estimations was first transformed to an ultrametric tree using the nnls method with phytools [75]. We also evaluated the diversity and phylodiversity partitioning into individuals ($\alpha$), species, and total ($\Upsilon$) components to visualize the mean number of ASVs and of lineages per component, and to estimate the effective number of individuals ($\beta_1$) and the effective number of species ($\beta_2$) in the system [74].

To assess statistically the abundance of taxa that differentiate the gut microbiota of each primate species and the hybrid individuals, we performed an Analysis of Composition of Microbiomes with Bias Correction (Ancom-bc) using the ANCOMBC package in R [76]. This method estimates the unknown sampling fractions and corrects the bias introduced by their differences among samples. We chose this method because it controls false discovery rates (FDR) while maintaining adequate power compared with other popular methods; it also provides p values with standard errors and confidence intervals for the differential abundance per taxon [76].

Finally, to assess the general association between the genetics of the host and its gut microbiota, we performed a Mantel test between the genetic distance and the microbiota dissimilarities of individuals of both *Alouatta* species. This analysis was based on microsatellite genotype data ([76]; Vázquez-Domínguez et al. [Unpublished]) for each of the 11 individuals of *A. palliata*, 10 *A. pigra* and nine hybrids. We ran the Mantel test in ade4 [77], based on a matrix of individual paired genetic distances estimated with Bruvo's genetic distance [78] with the poppr package [79] and the individual paired Unifrac unweighted distance matrix. This analysis was not done for *A. geoffroyi* due to lack of microsatellite data.

## Results

Sequencing yielded a total of 7,224,123 reads for the 16S rRNA V4 hypervariable region. Average number of reads per individual was 180,603. The denoising statistics after quality filtering (number of reads that were discarded during processing due to chimeras, etc.) for each sample are described in S2 Table. All 40 individual samples were included in the analyses since they reached the asymptote in the accumulation curves (S1 Fig). In the unfiltered dataset the majority (99.5%) of reads were assigned to Bacteria and 0.46% to Archaea. Two ASVs were assigned to Eukarya present in low abundances in a few individuals, two genera of Parabasalia: *Tetratrichomonas* found in *Alouatta palliata* and *Ateles geoffroyi,* and *Hypotrichomonas* only found in *A. geoffroyi.* The remaining reads (0.04%) were unassigned.

As expected for the mammalian gut microbiota, the most abundant Bacteria phyla were Bacillota (44.9%) and Bacteroidota (43.2%), surprisingly followed by Melainabacteria (6.1%), a non-photosynthetic sister group of Cyanobacteria, and Verrucomicrobiota (2.5%) (Fig 2a). The rest of phyla had an abundance of less than one percent. The proportion of the most abundant phyla varied both among host species and individuals. *Alouatta*'s gut microbiota showed a higher proportion of Bacillota over Bacteroidota, but *Ateles* had the opposite pattern (Fig 2a). This was reflected in the higher diversity of Bacillota families in *Alouatta*, particularly of Lachnospiraceae, Oscillospiraceae and Clostridiaceae, in contrast with the lower diversity of families in *Ateles*, with a major prevalence of Ruminococcaceae (genus *Faecalibacterium*) (Fig 2b; S2a Fig). On the other hand, the higher diversity of Bacteroidota in *Ateles* included various species of Prevotellaceae. Finally, the Bacteroidota community for *Alouatta* differed between the two species (S2b Fig).

Venn-diagrams for overall diversity, based on the unfiltered dataset, showed that the core gut microbiota of the three species comprises 106 ASVs (Fig 3a; S3 Table). Also, that *A. geoffroyi* has the most diverse and unique gut microbiota with 173 exclusive ASVs, followed by *A. palliata* (59), *A. pigra* (20) and hybrids (18). This general pattern was also observed for Bacillota, Bacteroidota and Verrucomicrobiota (Fig 3b, c, e). Notably, hybrids had more unique Bacillota ASVs than the parental species (Fig 4b). Although Melainabacteria was the third most abundant bacteria phylum overall, only six ASVs were present, two in the three host primates, one shared between *A. pigra* and *A. geoffroyi*, and three exclusively in *A. geoffroyi* (Fig 3d). Nine ASVs of Archaea were identified, three of which were shared between the three host species, while *A. geoffroyi* was the only with two unique ones (Fig 3f).

After filtering genera by a > 5% abundance, the bacterial community showed noticeable differences (Fig 2b). Of a total of 22 taxa, only six were shared among the three host species and the hybrids: *Prevotella, Gastranaerophilales, Bacteria,* Prevotellaceae_UCG-001, *Alloprevotella*, and *Faecalibacterium*, but with different abundances. Six abundant taxa were only present in howlers: Rikenellaceae_RC9_gut_group, *Cerasicoccus,* UCG-005, Clostridia_vadinBB60_ group, Clostridia_UCG-014, and *Anaeroplasma*. In addition, each howler species harbored two unique abundant taxa: *A. palliata* had *Treponema* and UCG-004, and *A. pigra* p-251-o5 and UCG-010. These four taxa were also present in the hybrid individuals. The only abundant taxon shared between the three species but not present in howler hybrids was Prevotellaceae_NK3B31_group. The howler hybrids exhibited three exclusive abundant taxa: *Phascolarctobacterium, Ruminococcus*, and uncultured, while they shared only Muribaculaceae with *A. palliata* and Lachnospiraceae_NK4A136_group with *A. pigra*. Based on this filtered dataset, the spider monkeys did not present unique abundant taxa but had major presence of some Prevotellaceae (Fig 2b).

Principal coordinate analysis (PCoA) for the dissimilarities between beta diversity based on unweighted Unifrac distances showed a clear differentiation of the *Ateles*'s microbiota

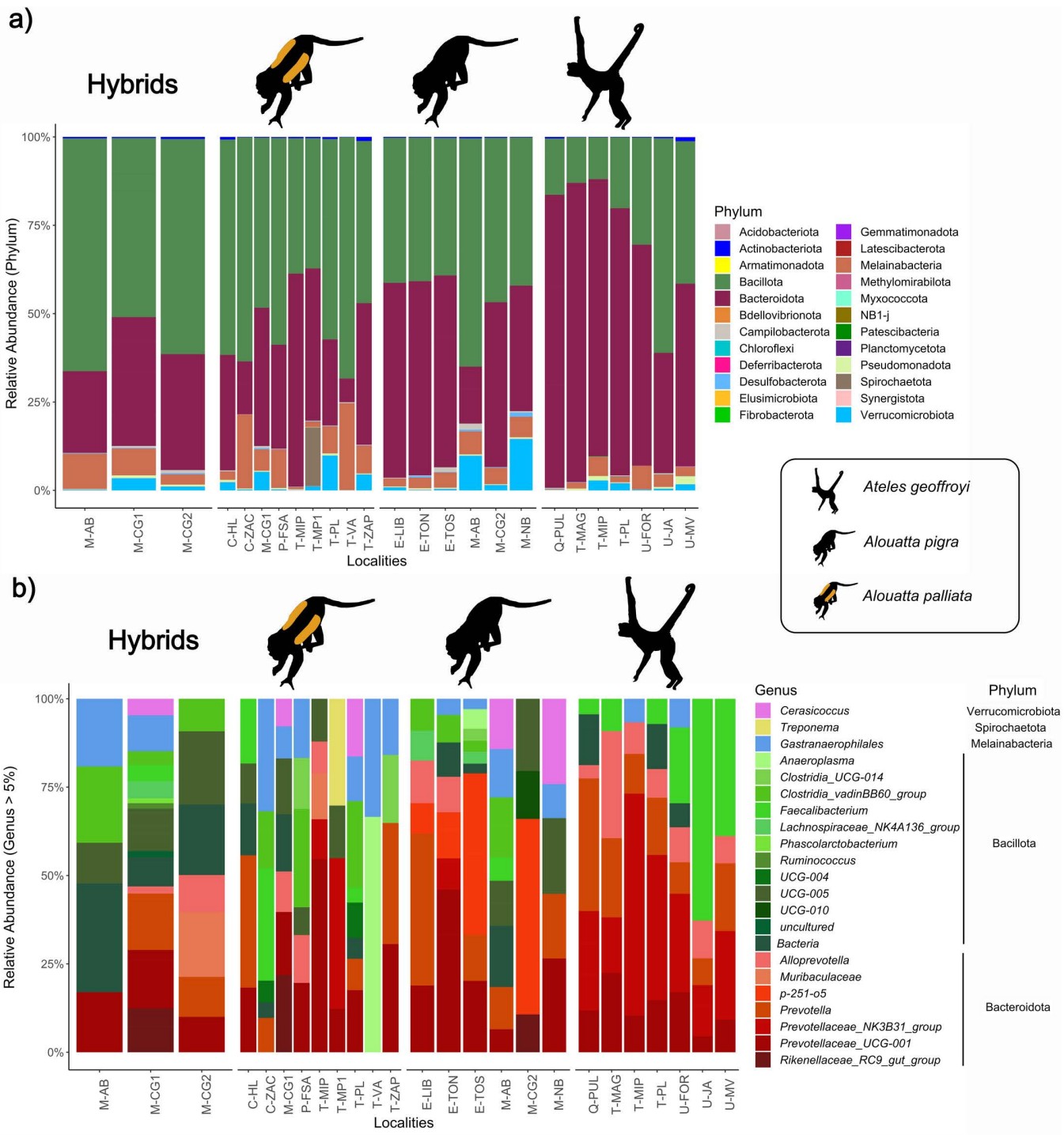

**Fig 2. Relative abundance graphs representing diversity of gut bacteria from wild populations of the three Mexican primates: spider monkey (*Ateles geoffroyi*), mantled howler monkey (*Alouatta palliata*), black howler monkey (*A. pigra*), and howlers hybrid individuals.** (a) Phylum composition and (**b**) genus composition filtered by > 5% abundance. Each bar represents a sampled locality (see S1 Table for locality names) grouped by species. The different taxa of bacteria are depicted by different colors, with corresponding names shown on the right. Black monkey silhouettes obtained from the open source, freely available at https://www.phylopic.org/images/aceb287d-84cf-46f1-868c-4797c4ac54a8/ateles-fusciceps; and https://www.phylopic.org/images/564c9708-bedb-4c1c-afc7-307f416901f0/alouatta-caraya.

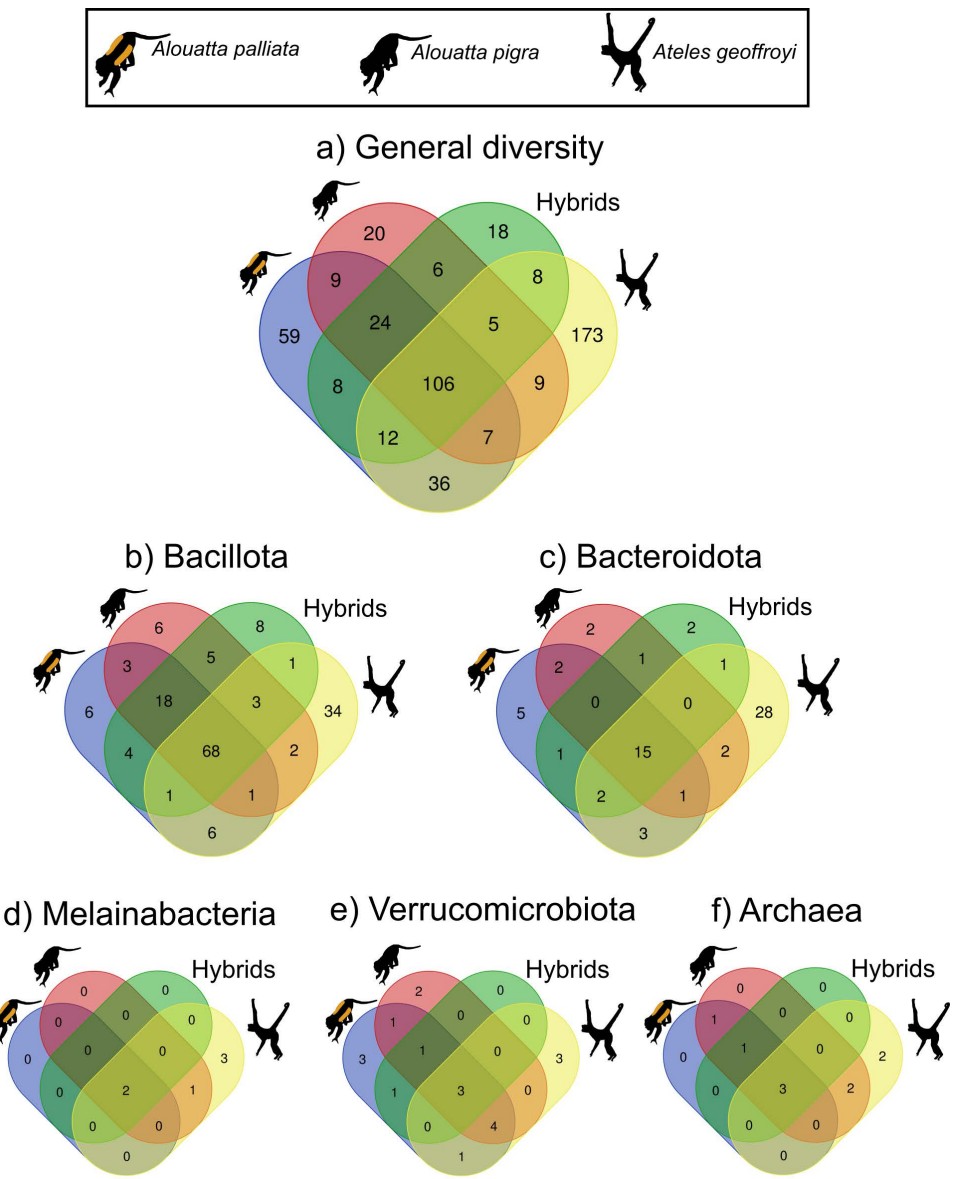

**Fig 3. Venn diagrams showing the number of unique and shared ASVs of the gut microbiota of the three Mexican primates: spider monkey (*Ateles geoffroyi*), mantled howler monkey (*Alouatta palliata*), black howler monkey (*A. pigra*); and the *Alouatta* hybrids.** (**a**) global diversity; (**b**-**e**) ASVs of the most abundant bacteria phyla; (f) ASVs of Archaea. Black monkey silhouettes obtained from the open source, freely available at at https://www.phylopic.org/images/aceb287d-84cf-46f1-868c-4797c4ac54a8/ateles-fusciceps; and https://www.phylopic.org/images/564c9708-bedb-4c1c-afc7-307f416901f0/alouatta-caraya.

(Fig 4). Most differences are reflected by axis 1 that accounted for 20.1% of variation. These differences were supported by Permanova results that showed significant statistical differences between *A. geoffroyi* and the two *Alouatta* species (Table 1a). On the other hand, the howlers' microbiota was more similar between them, yet differences were shown by axis 2 (7.8% of variation), also statistically significant (Table 1a). Hybrid individuals showed an intermediate diversity, albeit not significant, in relation with the parental species (Fig 3a; Table 1a); although they share more of their gut microbiota with *A. pigra* than with *A. palliata*. Overall

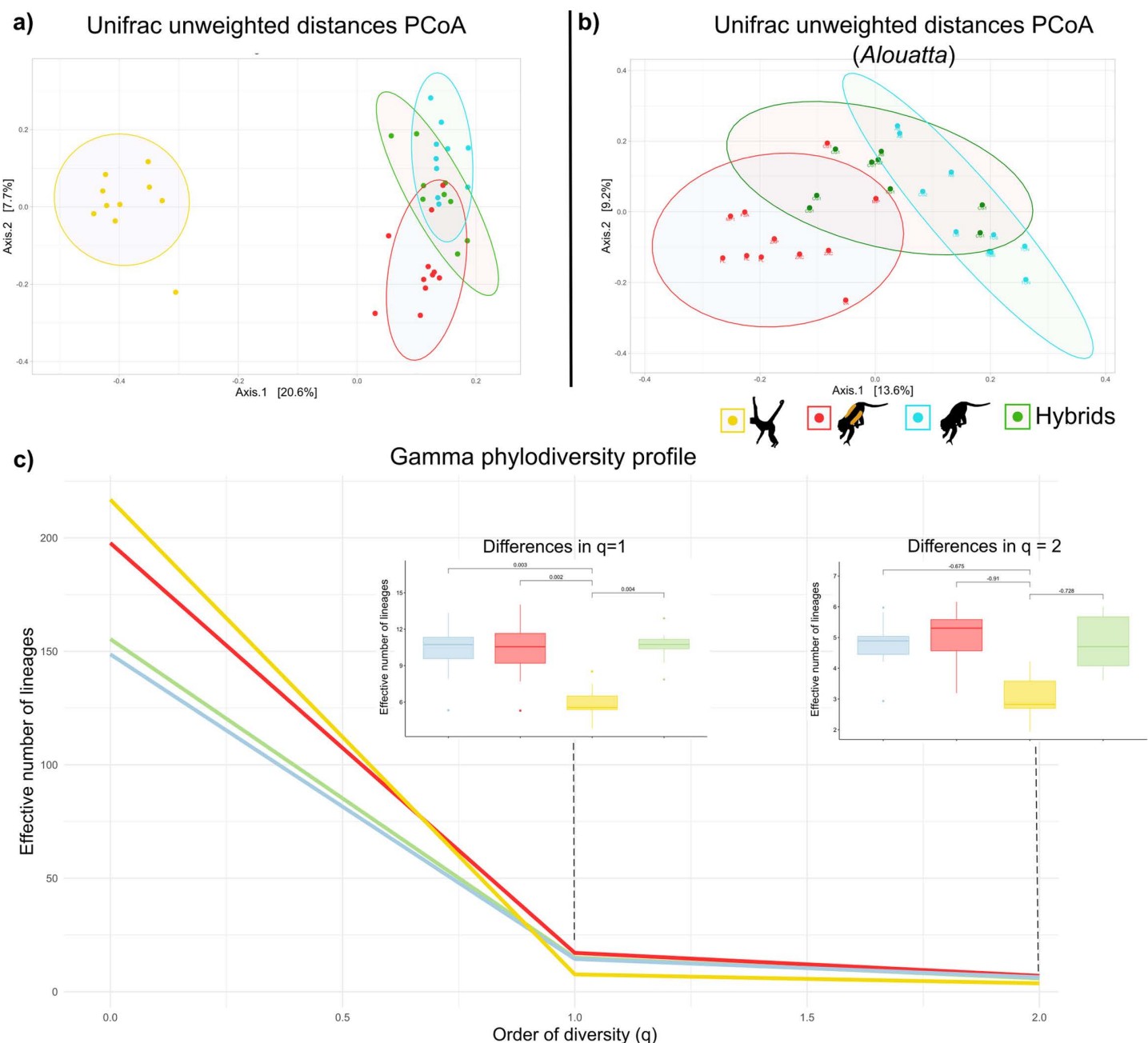

**Fig 4. Gut microbiota diversity of wild populations of the three species of Mexican primates: spider monkey (*Ateles geoffroyi*) individual samples indicated with yellow circles, mantled howler monkey (*Alouatta palliata*) in red, black howler monkey (*A. pigra*) in blue; *Alouatta* hybrid individuals in green.** Principal coordinate analysis (PCoA) based on Unifrac unweighted PCoA dissimilarities (**a**) for the three host species and hybrids and (**b**) for the *Alouatta* species and their hybrids, indicating the geographic locality. (c) Gamma phylodiversity profile based on Hill numbers with q values from 0 to 2 for each species. Significant statistical differences among species are shown in the box graphs for q = 1 (insert left graph) and q = 2 (right). Black monkey silhouettes obtained from the open source, freely available at https://www.phylopic.org/images/aceb287d-84cf-46f1-868c-4797c4ac54a8/ateles-fusciceps; and https://www.phylopic.org/images/564c9708-bedb-4c1c-afc7-307f416901f0/alouatta-caraya.

**Table 1. Permanova test results for the (a) Unifrac unweighted and (b) weighted distances of the gut microbiota between the three Mexican primate host species (*Alouatta pigra*, *A. palliata*, *Ateles geoffroyi*) and *Alouatta* hybrid individuals.**

| | Sums of Squares | F. model | R² | *p* value | Adjusted *p* value |
|---|---|---|---|---|---|
| **a** | | | | | |
| *A. pigra vs A. palliata* | 0.008504 | 5.133656 | 0.212717 | 0.001 | 0.006* |
| *A. pigra vs A. geoffroyi* | 0.096864 | 72.66019 | 0.801456 | 0.001 | 0.006* |
| *A. pigra vs A.* hybrids | 0.003754 | 2.667644 | 0.135636 | 0.010 | 0.060 |
| *A. palliata vs A. geoffroyi* | 0.084213 | 52.10614 | 0.732793 | 0.001 | 0.006* |
| *A. palliata vs A.* hybrids | 0.004735 | 2.782054 | 0.133868 | 0.008 | 0.048 |
| *A. geoffroyi vs A.* hybrids | 0.091187 | 66.94094 | 0.797476 | 0.001 | 0.006* |
| **b** | | | | | |
| *A. pigra vs A. palliata* | 0.073512 | 4.825081 | 0.202521 | 0.020 | 0.120 |
| *A. pigra vs A. geoffroyi* | 0.061216 | 11.75549 | 0.395069 | 0.001 | 0.006* |
| *A. pigra vs A.* hybrids | 0.021732 | 2.343139 | 0.121135 | 0.083 | 0.498 |
| *A. palliata vs A. geoffroyi* | 0.179010 | 12.60354 | 0.398801 | 0.001 | 0.006* |
| *A. palliata vs A.* hybrids | 0.019527 | 1.053038 | 0.055268 | 0.327 | 1.000 |
| *A. geoffroyi vs A.* hybrids | 0.089209 | 10.98499 | 0.392531 | 0.001 | 0.006* |

*Statistically different.

*Alouatta* results were concordant with geographic location (Fig 4b). PCoA of weighted Unifrac distances reflected less variation among the microbiota of the three species (S3 Fig), with statistical differences between spider monkeys and the howlers (Table 1b); this could be related to the diversity of rare taxa present in *A. geoffroyi*. Patterns did not differ for the species or the hybrids when based on the rarefied dataset (S4 Fig).

The gamma phylodiversity and gamma diversity profiles based on Hill numbers (Fig 4c, S5 Fig), when considering rare taxa and not taking abundance into account (q = 0), showed that *A. geoffroyi* had the highest counts of effective lineages (211.4) and of effective ASVs (1340), followed by *A. palliata* (192.15 lineages; 1212 ASVs) and last *A. pigra* (145.01 lineages; 1030 ASVs). However, when abundance is considered (q > 0), *A. geoffroyi* was the least diverse, differences that were statistically significant (q = 1 and q = 2; Fig 3c).

Hierarchical diversity partitioning showed that each species had an average of 1153.75 ASVs in their gut microbiota, although communities were dominated by few taxa (ca. 45-132 ASVs, depending on q value) (Table 2a). A similar pattern is present in the phylodiversity partitioning but with lower values (Table 2b). The effective number of individuals ($\beta_1$) was *ca.* 3 (2.68-3.68, depending on q value) and the effective number of species ($\beta_2$) was *ca.* 2 (2.11-2.54) (Table 2a). When considering the phylogenetic relationships, the global diversity ($\Upsilon$) was similar to the species diversity (Table 2b), suggesting that the gut microbiota of each species is highly related to that of the other species.

Regarding the assessment of the abundance of taxa that differentiate the gut microbiota of each primate species and the hybrid individuals, Ancom-bc results showed that each primate species has a particular overrepresented community; comparatively, hybrid individuals did not have specific overrepresented taxa. *Alouatta pigra* had the most overrepresented bacteria taxa, emphasizing p-251-o5, Akkermansiaceae, and Clostridia_UCG_014. The Clostridia_vadinBB60_group, Erysipelatoclostridiaceae, and Comamonadaceae were particularly represented in *A. palliata*, while Acidaminococcaceae, Bacteria_6, and Tannerellaceae were so between *Alouatta* species. The most overrepresented taxa in *Ateles* were Selenomonadaceae, Succinivibrionaceae, and Xanthomonadaceae families (Fig 5).

**Table 2. Diversity based on Hill numbers of the gut microbiota of wild populations of the Mexican primates partitioned by individual (α), species, and global (ϒ).** Values indicate the hierarchical partitioning of (a) diversity and (b) phylodiversity. Numbers in black represent the average number of effective (a) ASVs and (b) lineages in terms of q values (q = 0 richness or number of taxa/lineages, q = 1 abundant taxa/lineages, q = 2 dominant taxa/lineages; [74]), while numbers in gray depict the effective number of individuals (β₁) and effective number of species (β₂).

| | q = 0. | q = 1. | q = 2 |
|---|---|---|---|
| **a** | | | |
| Individual diversity (α) | 341.45 | 49.308 | 14.964 |
| β1 | 3.379 | 2.676 | 3.002 |
| Species diversity | 1153.75 | 131.968 | 44.927 |
| β2 | 2.528 | 2.110 | 2.540 |
| Global diversity (ϒ) | 2917 | 278.948 | 114.134 |
| **b** | | | |
| Individual phylodiversity(α) | 76.251 | 8.246 | 3.702 |
| β₁ | 4.445 | 1.805 | 1.479 |
| Species phylodiversity | 338.958 | 14.883 | 5.476 |
| β₂ | 1.000 | 1.004 | 1.004 |
| Global phylodiversity (ϒ) | 338.958 | 14.938 | 5.499 |

The Mantel tests showed a significant positive correlation (r = 0.303; $p$ = 0.0001) between host genetic distances and gut microbial dissimilarities across all howler individuals (S6 Fig).

## Discussion

### Diversity of the gut microbiota of the three Mexican primate species

The study of the gut microbiome of non-human primates (NHP) enhances our understanding of their ecology and evolution. We evaluated interspecific and intraspecific patterns of diversity, abundance and composition of gut microbial communities of wild populations of the three Mexican Atelidae primates from different regions across their distribution in Mexico and encompassing a hybrid zone. Our findings enhance the information about gut microbiota of wild populations of atelids, and specifically regarding their northernmost populations in America.

The most abundant taxa in the studied primate populations were concordant with that reported for other primates, including humans [80], like Prevotellaceae (i.e., *Prevotella*), Ruminococcaceae (i.e., *Faecalibacterium*), Clostridiaceae, Lachnospiraceae and Oscillospiraceae. Diversity of the gut microbiota at the phylum level was similar to what has been reported in wild and captive atelids [40,50,53,81], showing a higher influence of phylogenetic than environmental factors. Our findings show a few ASVs assigned to Melainabacteria with a significant abundance in most samples, a phylum related to Cyanobacteria that has been identified as a member of the gut microbiota in mammals [82]. Moreover, Melainabacteria, particularly *Gastranaerophilales*, have been suggested to inhabit low-oxygen environments like the gut and are associated to vitamin synthesis and fermentative metabolism [83]. They seem to be an important part of the gut microbial community in atelids, present at low diversity yet in relatively high abundances in most genera except *Lagothrix* [40,50,53,81]. Notably, we found a greater presence of Verrucomicrobiota and less abundance of Pseudomonadota and Tenericutes compared with other studies in the family Atelidae [40,50,53,81].

Consistent with phylosymbiosis, differences in the gut microbiota diversity of the three primate species were significantly associated with host phylogeny; hence, with host physiology and behavior. The host evolutionary history is also known to have strong effects

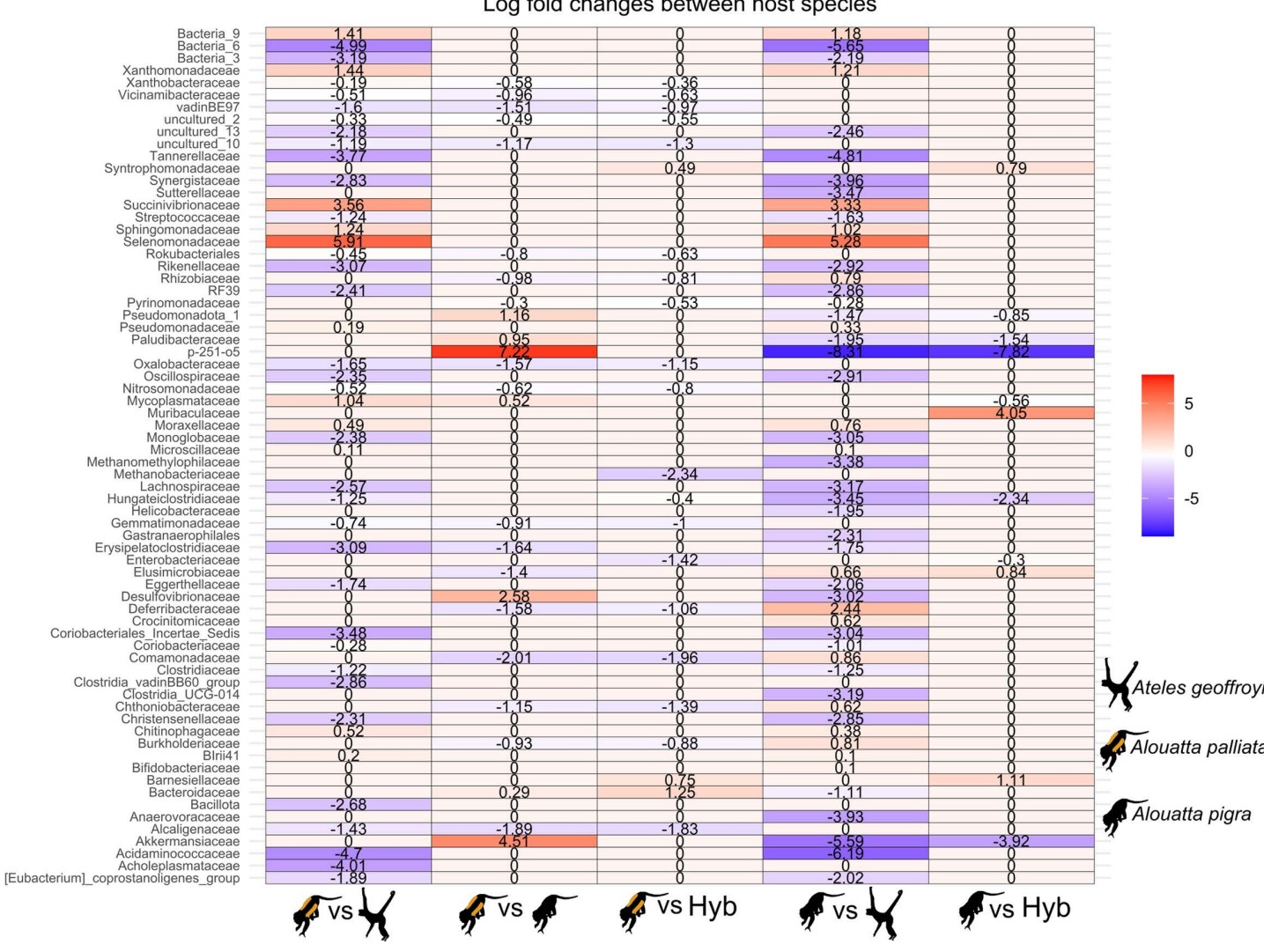

**Fig 5. Differential abundance of the gut microbiota of the three Mexican primates (*Alouatta palliata*, *A. pigra*, *Ateles geoffroyi*) and *Alouatta* hybrids.** Log fold changes of the microbial communities at family level calculated with the Ancom-bc analysis between paired host species. The color gradient (scale on the right) indicates the overrepresented taxa in red and the underrepresented taxa in blue. Black monkey silhouettes obtained from the open source, freely available at https://www.phylopic.org/images/aceb287d-84cf-46f1-868c-4797c4ac54a8/ateles-fusciceps; and https://www.phylopic.org/images/564c9708-bedb-4c1c-afc7-307f416901f0/alouatta-caraya.

on the gut microbiota in other primates [18–21]. *Ateles geoffroyi* had the most differentiated gut microbiota, harboring more rare taxa and significant diversity and composition contrasts (Figs 2-4). In addition, the time since host clade divergence has been shown to be positively associated to the level of discrimination of the microbial community between species within the host clades [11]. Therefore, the phylogenetic patterns we found agree with the long divergence time (15-16 Mya) between *Ateles* and *Alouatta* [57,58]. Differences in the gut microbiota between the howler species were less strong, consistent with previous studies [51], although with distinct diversity patterns. This might be related, among others, with time-space differences, including the wider distribution and concurrent environmental (e.g., vegetation type) variation of our study; and also geographic

distribution, since the known information for *A. palliata* comes from populations in Costa Rica and Nicaragua [51]. Indeed, the more extensive sampling of the howler monkeys within their native distribution in Mexico and the fact that we included the spider monkey in our study, provides further information for the understanding of the mechanisms driving these host taxonomic differences.

Life history variances and distinct feeding behaviors must also have an impact on gut microbial diversity; however, like in other studies with NHPs, correlation between diet and phylogeny complicates disentangling the factors directly affecting gut microbiota [22,33,38]. *Ateles* is mostly frugivore and *Alouatta* mainly folivore [41]. The higher proportion of Bacillota over Bacteroidetes present in *Alouatta* implies a better fermentation efficiency [84]. Indeed, howlers are adapted to a folivore diet which, as hind-gut fermenters, present large cecum and colon [85], slow food passage rates [41], and other physiological adaptations [86,87]. Accordingly, *Alouatta* microbial communities had more abundant and enriched taxa that have potential fermentative effects associated with folivory (i.e., Clostridiaceae) [22], while *Ateles* had a major diversity and abundance of Ruminococcaceae and Prevotellaceae, associated with degradation of sugars like those of fruits [17,21]. These patterns of gut microbial community composition were also present where *A. geoffroyi* and *A. palliata* individuals are sympatric (e.g., Playa and Mirador Pilapa localities at Los Tuxtlas region; Fig 1). Importantly, *Bifidobacterium* was previously described as a central agent in gut microbiota communities in *Alouatta* [51,88], but in the populations we studied it was only present in *A. geoffroyi*, albeit in low abundances. Altogether, these results show further evidence of both the significant effect of phylogeny on the microbial diversity and the close link between host physiology and gut microbiota in atelids.

Interesting patterns were observed regarding the gut microbiota of hybrid individuals. They exhibit some unique taxa, a higher diversity of Bacillota compared to the parental species, and an intermediate overall diversity between the parental species, yet more similar to *A. pigra* (Fig 4). The latter is likely associated with the fact that only female hybrids are viable in these howler monkeys, resulting from the reproduction between *A. palliata* males and *A. pigra* females [54]. Hence, *A. pigra* mothers transfer their microbiome to their offspring and, therefore, the observed higher similarity of gut microbiota of hybrids to *A. pigra*. Some of the unique taxa have been previously reported in human guts (i.e., *Colidextribacter* and *Eisenbergiella*) [89,90], others are potentially beneficial fermenters (i.e., uncultured Lachnospiraceae and Clostridiaceae, *Lactobacillus*, *Pelosinus* and *Bacteroides ovatus*) [45,91,92], and some are potential pathogens (i.e., *Bilophila*, *Mogibacterium* and *Alistipes*) [93–95]; but all these are present in low abundances in our samples. The gut microbial community composition in the region of sympatry clustered by locality in the PCoA regardless of host species, showing a potential effect of the environment and geographic location at this phylogenetic level [25,96]. In addition, the hybrid individual sampled in AB, a locality encompassing more continuous and better conserved tall tropical evergreen forest, showed a decrease abundance of *Faecalibacterium, Prevotella* and *Cerasicoccus*, which are potential fermentative bacteria [44]. On the other hand, hybrids in CG1 and CG2, where the habitat is more fragmented and disturbed, showed more abundant taxa in comparison to the parental species. This could be related to different dietary patterns of individuals among localities with bacterial communities compensating for these changes, as has been demonstrated between seasons [35], or associated to disturbance dynamics of the environment [97]. Finally, we found a significant effect of host genetics across all howler individuals, where genetically closer individuals have a more similar microbiota. Altogether, these results suggest that both environmental and genetic factors affect the diversity, abundance and composition of the gut microbiota in *Alouatta* and their hybrids.

## Environment, geography and gut microbial communities at the intraspecific level

Intraspecific differences in the gut microbial communities were more subtle, with vegetation type and geography as possible influencing factors. *Ateles geoffroyi* showed distinct gut microbial communities in individuals inhabiting different regions/localities; specifically, those from Los Tuxtlas and Uxpanapa regions, which are characterized mostly by tall evergreen tropical forest, have a major abundance of *Gastranaerophilales* (MIP and FOR localities) and also present communities dominated by the abundant Bacteroidetes (MV, PLAG, FOR); in addition, MIP is the only locality where individuals harbor a considerable abundance of Verrucomicrobiota but low presence of *Faecalibacterium*. Lastly, a notable increment in Bacillota abundance, specifically Ruminococcaceae, was shown by individuals from the Uxpanapa region (FOR, JA, MV). These patterns indicate that environmental and geographic factors likely affect the community composition of the gut microbiota in *A. geoffroyi*, although with lower magnitude compared to host phylogeny. Furthermore, *A. geoffroyi* showed across all samples a gut microbiota with enriched Selenomonadaceae (present in the rumen of cows; [98]), comparable to previous reports for captive individuals [52,53]. This could be associated with the degree of habitat modification and fragmentation, along with fruit availability, but it is something that needs further study.

Vegetation type, anthropogenic habitat modification and geographic isolation are likely associated with the intraspecific differences in gut microbiota shown by *A. palliata*. The individual sampled from Montepío (MP1) in Los Tuxtlas region was the only of all *A. palliata* individuals harboring *Treponema*, and in considerable abundance. Despite the genus *Treponema* appears to be a key component of the primate-wide gut microbiota, some species can be pathogens like *T. pallidum* [99,100]. This region is the northernmost distribution of *A. palliata* and MP1 is a forest fragment next to a small coastal village, in which howler monkeys live near humans and domestic animals. It is then important to extend surveys in this region to follow up of the potentially occurrence of such pathogenic bacteria. Other factors like geographic location, which influences vegetation type and consequently diet, could be associated with differences in the microbial communities of this species like it has been documented in *A. pigra* [17,35,44], as well as sex, age and temporal differences in sampling. In fact, we identified different composition and diversity patterns in *A. pigra* individuals from the Escárcega region (El Tormento localities) in comparison with those previously reported [50,51]. Again, this may be due to temporal sampling, while our smaller sample size should also be considered. Additionally, our results show differences in microbial composition between close localities within regions (TON-TOS; CG2-NB), which can result from anthropogenic barriers like heavy traffic roads or farmlands hindering the movement of howler individuals between groups, but also due to differences associated to social groups and social behavior [46].

The three species of primates in Mexico are endangered, with declining populations and significant threats, including hunting, illegal trade and habitat loss and fragmentation [101–103]. One of the main causes of dead of trafficked monkeys and of NHP individuals living in ex-situ conditions is gut affections related to gut microbiota [104]. Hence, understanding the host-gut microbiome interactions in NHP is crucial for the conservation of these species [2,27]. To mention just a few purposes, individual gut microbiota information could aid in the organization and translocation success of individuals for conservation purposes [105], and to develop probiotics for health maintenance of ex-situ populations [88]. Future research in the study of host-microbiome associations requires large scale studies with cross sectional samples, aiming to cover the distribution of host species in more detail and including more populations and sympatric individuals of different species. Studies should target to

obtain detailed information related to the host at the time of sampling (e.g., environmental variables, vegetation type, forest cover, food availability), which will enable us to continue deciphering ecological and evolutionary factors affecting the gut microbiota. Furthermore, in terms of intraspecific variation in NHP gut microbial communities, we should perform studies based on longitudinal sampling of microbiota associated to host genetics and other environmental and landscape variables.

## Supporting information

**S1 Fig.  Accumulation curves for the 16S rDNA V4 sequences by host species.**
(PDF)

**S2 Fig.  Relative abundance graphs of the two most abundant Phyla of gut bacteria.**
(PDF)

**S3 Fig.  Principal coordinate analysis (PCoA) based on weighted Unifrac distance dissimilarities.**
(PDF)

**S4 Fig.  Principal coordinate analysis (PCoA) based on Unifrac distance dissimilarities based on rarefied data.**
(PDF)

**S5 Fig.  Gamma diversity based on Hill numbers.**
(PDF)

**S6 Fig.  Mantel test graph results for the correlation between gut microbiota dissimilarities and genetic dissimilarities.**
(PDF)

**S1 Table.  Sampling localities and vegetation type for the wild Mexican primate species studied** .
(PDF)

**S2 Table.  Denoising statistics after quality filtering for each sample** .
(PDF)

**S3 Table.  Amplicon sequence variants (ASVs) identified per host species** .
(PDF)

## Acknowledgements

We thank Samuel Cardós, Ruben Mateo, Kyle Shaney and Pablo Gutiérrez for their help and support during fieldwork. Authorities from Hacienda La Luz, Centro de Investigación y Transferencia de Tecnología Forestal 'El Tormento', Parque Arqueológico Comalcalco, and Natural Protected Areas kindly granted permission for fieldwork and feces sample collection. Special thanks to Mariana Vázquez-Reyes for her participation in field work and microsatellite amplification, Gabriela Borja-Martínez and Miguel Baltazar for computational assistance, Nancy Gálvez Reyes and Osiris Gaona for laboratory support and Graciela García Guzmán for her aid in obtaining the sampling permit. Open access was obtained thanks to the UNAM agreement that covers article publication charges. D.Z. acknowledges that this paper was a part of his Master's thesis in the Posgrado en Ciencias Biológicas de la Universidad Nacional Autónoma de México (UNAM).

## Author contributions

**Conceptualization:** Diego Zubillaga-Martín, Brenda Solórzano-García, Luisa I. Falcón, Ella Vázquez-Domínguez.

**Formal analysis:** Diego Zubillaga-Martín, Alfredo Yanez-Montalvo, Arit de León-Lorenzana.

**Funding acquisition:** Ella Vázquez-Domínguez.

**Investigation:** Diego Zubillaga-Martín, Brenda Solórzano-García, Ella Vázquez-Domínguez.

**Methodology:** Diego Zubillaga-Martín, Brenda Solórzano-García, Alfredo Yanez-Montalvo, Arit de León-Lorenzana, Luisa I. Falcón, Ella Vázquez-Domínguez.

**Writing – original draft:** Diego Zubillaga-Martín, Ella Vázquez-Domínguez.

**Writing – review & editing:** Diego Zubillaga-Martín, Brenda Solórzano-García, Alfredo Yanez-Montalvo, Arit de León-Lorenzana, Luisa I. Falcón, Ella Vázquez-Domínguez.

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
