## [Decision Letter · Decision Letter 0]

24 Oct 2024

PONE-D-24-38914Gut microbiota signatures of the three Mexican primate species, including hybrid populationsPLOS ONE

Dear Dr. Vázquez-Domínguez,

Thank you for submitting your manuscript to PLOS ONE. After careful consideration, we feel that it has merit but does not fully meet PLOS ONE’s publication criteria as it currently stands. Therefore, we invite you to submit a revised version of the manuscript that addresses the points raised during the review process.

 Both reviewers agreed that this study provides valuable information for the research community. However, both noted a number of issues regarding experimental setup, data analysis, interpretation, and discussion that need to be adequately addressed before further consideration can be made.

We look forward to receiving your revised manuscript.

Kind regards,

Brenda A Wilson, Ph.D.

Academic Editor

PLOS ONE

Journal Requirements:

“This project was funded by Programa de Apoyo a Proyectos de Investigación e Innovación Tecnológica-Dirección General de Asuntos del Personal Académico (PAPIIT-DGAPA, UNAM; Projects IN202122 and IN202819) awarded to E.V.D., including a Postdoctoral scholarship from DGAPA to B.S.G. D.Z. was supported by a graduate scholarship from CONAHCyT (1146867).”

3. Please expand the acronym “CONAHCyT” (as indicated in your financial disclosure) so that it states the name of your funders in full.

4. We note that you have referenced (unpublished data; Vázquez-Domínguez et al. in prep) on page 7, which has currently not yet been accepted for publication. Please remove this from your References and amend this to state in the body of your manuscript: (ie “Bewick et al. [Unpublished]”) as detailed online in our guide for authors

5. We note that [Figure 1] in your submission contain [map/satellite] images which may be copyrighted. All PLOS content is published under the Creative Commons Attribution License (CC BY 4.0), which means that the manuscript, images, and Supporting Information files will be freely available online, and any third party is permitted to access, download, copy, distribute, and use these materials in any way, even commercially, with proper attribution. For these reasons, we cannot publish previously copyrighted maps or satellite images created using proprietary data, such as Google software (Google Maps, Street View, and Earth). For more information, see our copyright guidelines: http://journals.plos.org/plosone/s/licenses-and-copyright.

Reviewers' comments:

Reviewer's Responses to Questions

**Comments to the Author**

1. Is the manuscript technically sound, and do the data support the conclusions?

Reviewer #1: Yes

Reviewer #2: Partly

2. Has the statistical analysis been performed appropriately and rigorously? 

Reviewer #1: Yes

Reviewer #2: No

3. Have the authors made all data underlying the findings in their manuscript fully available?

Reviewer #1: No

Reviewer #2: No

4. Is the manuscript presented in an intelligible fashion and written in standard English?

Reviewer #1: Yes

Reviewer #2: Yes

5. Review Comments to the Author

Reviewer #1: 

Summary: The study presented in “Gut microbiota signatures of the three Mexican primate species, including hybrid populations” by Zubillaga and colleagues examines the gut microbiome composition of wild populations of the three Mexican primates and hybrid populations. Their results revealed distinct microbiome compositions in the 3 NHP species strongly shaped by host specific factors.

While I believe this is an important work and is a good fit for the PLOS ONE journal, there are number of minor concerns I would like the authors to address.

Minor Concerns:

Sample size: A total sample size of 40 with 9-11 samples might reduce the power the statistical analysis. Did the authors do a power analysis to determine if these samples sizes are sufficient?

Rarefaction: This is a crucial step in microbiome studies which addresses biases that arise from unequal sampling depths allowing for fair comparisons between samples. Given that all 40 samples had more than 20,000 reads, why did the authors choose not to perform rarefaction. With such high number of reads per sample, they would not have to drop any samples even with rarefaction. Could the authors check how rarefaction would impact the results?

Reviewer #2: 

In this brief report the authors provide results from a study of gut microbiota from three wild primate populations of black and mantled howler monkeys (Alouatta palliata and Alouatta pigra, respectively) and spider monkeys (Ateles geoffroyi) from different regions in Mexico. They also examined hybrid offspring between the groups. The methodology used was genomic DNA extraction, followed by 16S rRNA gene amplification and sequencing of the V4 region using Illumina MiSeq platform. The resulting reads were processed into ASVs with QIIME2, followed by SILVA classification and alignment with MAFFT and FastTree, phyloseq, rarefaction, and PCoA analysis. The authors examined the interspecific and intraspecific gut microbial diversity, abundance, and composition profiles of 40 individuals from the three populations. They found that interspecific differences of gut microbial diversity were higher than intraspecific differences. They found that hybrids had microbial diversities in between that of their parental species, but also showed some unique microbial taxa. They also found evidence that microbial compositions were affected by environment, geographic distribution, and host genetics. Overall, the results are interesting and so long as the authors make the sequences available to the community through public sequence databases the data will contribute to our understanding of the microbial communities of primates. However, there some issues that if adequately addressed would significantly improve this manuscript:

1. There does not appear to have been any power analysis performed to assess the statistical significance of the results. Is a total sample size of 40 individuals with 3 species groups and several hybrid groups provide sufficient power? This is particularly a concern considering that the authors mention their results differ from other reported studies and that it might be due to sample size differences or due to temporal differences in sampling. Not knowing the more likely contributor to the differences lowers the overall impact of the study.

2. Line 253 – since the cited study is in preparation (not published), this data should be either included here in this manuscript or excluded completely. There is limited way for a reviewer to make an adequate assessment of the quality of the results described without that data.

3. Figure 3 is not very useful. A more comprehensive analysis of the connection of the taxa in each of the samples with the nature of the environment, location, food sources at time of sample collection, dietary preferences, and metabolic implications regarding the taxa present would have been more meaningful. For example, what taxa are shared between the hybrids and their parent groups? Does this make sense with the environmental factors and their diets or locations? More focus on the differences observed in taxa based on some of the demographics would be more informative.

4. The study design was limited with regard to the resolution of taxonomic classification based on PCR amplification of the V4 region. The authors should include information regarding the number of reads that were discarded during processing due to chimeras etc.

5. The authors note the similarities and differences among the samples, but they do not offer much in terms of explanation for the observed differences. Also, there is little comparison to what was previously reported for these primates.

6. It does not appear that the authors have submitted their sequencing data to a public database to share with the research community. This should be done.

6. PLOS authors have the option to publish the peer review history of their article (what does this mean? ). If published, this will include your full peer review and any attached files.

**Do you want your identity to be public for this peer review?** For information about this choice, including consent withdrawal, please see our Privacy Policy .

Reviewer #1: No

Reviewer #2: No

---

## [Author Response · Author response to Decision Letter 0]

10 Dec 2024

Response is included as a pdf document called "Response to reviewers"

---

## [Decision Letter · Decision Letter 1]

3 Jan 2025

Gut microbiota signatures of the three Mexican primate species, including hybrid populations

PONE-D-24-38914R1

Dear Dr. Vázquez-Domínguez,

We’re pleased to inform you that your manuscript has been judged scientifically suitable for publication and will be formally accepted for publication once it meets all outstanding technical requirements.

Kind regards,

Brenda A Wilson, Ph.D.

Academic Editor

PLOS ONE

Additional Editor Comments (optional):

Reviewers' comments:

Reviewer's Responses to Questions

**Comments to the Author**

1. If the authors have adequately addressed your comments raised in a previous round of review and you feel that this manuscript is now acceptable for publication, you may indicate that here to bypass the “Comments to the Author” section, enter your conflict of interest statement in the “Confidential to Editor” section, and submit your "Accept" recommendation.

Reviewer #1: All comments have been addressed

Reviewer #2: All comments have been addressed

2. Is the manuscript technically sound, and do the data support the conclusions?

Reviewer #1: Yes

Reviewer #2: Yes

3. Has the statistical analysis been performed appropriately and rigorously? 

Reviewer #1: Yes

Reviewer #2: Yes

4. Have the authors made all data underlying the findings in their manuscript fully available?

Reviewer #1: Yes

Reviewer #2: Yes

5. Is the manuscript presented in an intelligible fashion and written in standard English?

Reviewer #1: Yes

Reviewer #2: Yes

6. Review Comments to the Author

Reviewer #1: The authors addressed my concerns. I don't have any more concerns at this time. The paper can be accepted

Reviewer #2: (No Response)

7. PLOS authors have the option to publish the peer review history of their article (what does this mean? ). If published, this will include your full peer review and any attached files.

**Do you want your identity to be public for this peer review?** For information about this choice, including consent withdrawal, please see our Privacy Policy .

Reviewer #1: No

Reviewer #2: No

---

## [Editor Report · Acceptance letter]

PONE-D-24-38914R1

PLOS ONE

Dear Dr. Vázquez-Domínguez,

I'm pleased to inform you that your manuscript has been deemed suitable for publication in PLOS ONE. Congratulations! Your manuscript is now being handed over to our production team.

Kind regards,

on behalf of

Dr. Brenda A Wilson

Academic Editor

PLOS ONE